# Self-compassion and association with distress, depression, and anxiety among displaced Syrians: A population-based study

**Sarah Alsamman**[1], **Rana Dajani**[2,3], **Wael K. Al-Delaimy**[4]*

**1** University of California, San Diego School of Medicine, San Diego, California, United States of America, **2** Department of Biology and Biotechnology, The Hashemite University, Zarqa, Jordan, **3** MIT Refugee Action Hub (ReACT), Massachusetts Institute of Technology, Cambridge, Massachusetts, United States of America, **4** Herbert Wertheim School of Public Health and Human Longevity Science, University of California, San Diego, San Diego, California, United States of America

* waldelaimy@health.ucsd.edu

**Data Availability Statement:** All relevant data are within the manuscript and its Supporting

## Abstract

Displaced communities are at increased risk of poor mental health with limited resources for treatment. Self-compassion moderates the impacts of stressors on mental health in high-income country general population samples, but its impact has not been described among people who have experienced displacement and associated trauma. The aim of this study was to characterize the associations between self-compassion, mental health, and resilience in a sample of displaced Syrian adults living in Jordan. This is a cross-sectional study using four validated survey tools measuring self-compassion, resilience, mental health, and traumatic exposure. Syrian adults who presented to four different community organizations serving refugees within Amman, Jordan were invited to participate. A total of 272 displaced Syrians were included in the final analysis. A majority of those surveyed were positive for emotional distress (84.6%), depression (85.7%), and anxiety (76.5%). In univariate analysis there was a significant lower risk of emotional distress, depression, and anxiety, with both higher resilience and self-compassion. However, in the multivariate model only self-compassion remained significantly associated with less emotional distress, depression, and anxiety, independent of resilience and other covariates. Female gender, poor financial stability, and high levels of traumatic exposure were also identified as persistent predictors of mental health morbidity. The findings of this study suggest that self-compassion is associated with less distress, depression, and anxiety in displaced individuals; suggesting it might be protective against poor mental health. Self-compassion is a modifiable factor that can be utilized as a tool by healthcare professionals and communities caring for refugees to promote positive mental health outcomes.

## Introduction

More than a decade into the conflict, Syrians make up nearly one-third of displaced populations worldwide [1]. Since 2011, 6.8 million Syrians have been displaced, 5.6 million of whom

Information files. we removed all identifying
information.

**Funding:** This study was funded by the T. Denny
Sanford Institute for Empathy and Compassion at
UC San Diego to support SA to conduct the study.
The funders had no role in study design, data
collection and analysis, decision to publish, or
preparation of the manuscript.

**Competing interests:** The authors have declared
that no competing interests exist.

**Abbreviations:** PTSD, Post-traumatic stress
disorder; US, United States; TEC, Trauma
Experiences Checklist; HSCL, Hopkins Symptom
Checklist; CD-RISC, Connor-Davidson Resilience
Scale; SCS, Self-Compassion Scale—Short Form;
LMICs, Low- and middle-income countries; HICs,
High income countries.

are hosted in neighboring countries Turkey, Lebanon, Jordan, Iraq, and Egypt [1]. Refugees
are exposed to numerous stressors pre-migration in their country of origin and during transit
to a different country, including violence, torture, death of loved ones, and lack of healthcare
and basic resources [2]. Post-migration, during resettlement, they face new challenges includ-
ing discrimination, language barriers, and unemployment [2]. As a result of these stressors,
refugee populations have a well-documented higher mental health morbidity with increased
rates of post-traumatic stress disorder (PTSD), depression, and anxiety when compared to the
general population [3]. In a systematic review of studies on mental health of refugees of various
origins from 2003 to 2020 the prevalence of PTSD was 31.46%, the prevalence of depression
was 31.5%, and the prevalence of anxiety was 11% [3]. A recent systematic review of 15 studies
conducted in 10 different countries demonstrates Syrian refugees have a 10 times higher prev-
alence of PTSD (43%), anxiety (26%), and depression (40%) than the general population [4].
Understanding the factors that contribute to positive mental health despite suffering and
trauma can aid in the development of interventions to improve mental health outcomes in
these communities.

Resilience describes the individual positive response and adaptation to adversity. This con-
cept encompasses both individual skills and attributes as well as supportive social environ-
ments and family networks that work in unison to overcome traumatic experiences [5].
Amongst cancer patients, resilience is associated with psychological adjustment in the setting
of oncological treatment and its consequences [6]. Another study found that resilience is pro-
tective against postpartum depression in survivors of childhood trauma [7]. Some studies have
demonstrated that resilience is a protective factor against development of psychopathology in
refugees and recent immigrants, including Syrians [8, 9]. However, other studies report that
resilience is not predictive of distress or mental health symptoms after major life events or
traumatic experiences [10, 11]. Resilience is complex and multidimensional as it captures a set
of individual traits within a social environment [12]. Therefore, interventions that effectively
modulate resilience may act through other mechanisms like self-compassion to improve men-
tal health outcomes.

Self-compassion is defined as being kind and understanding towards oneself, perceiving
one's experiences as part of the larger human experience, and holding painful thoughts and
feelings in balanced awareness [13]. It has been shown to moderate the impact of stressors on
mental health [14]. Studies have demonstrated the role of self-compassion in reducing depres-
sion and anxiety in the context of various stress exposures including chronic illness and
domestic violence [14–17]. Additionally, self-compassion interventions in the general popula-
tion are shown to effectively increase self-compassion and improve mental health [18].
Amongst refugees, the literature on self-compassion and mental health is growing. In a sample
of Kurdish refugees resettled in Norway, self-compassion was associated with fewer symptoms
of depression [19]. In a study of Eritrean asylum seekers in the Middle East, self-compassion
was significantly modulated by a Mindfulness Based Trauma Recovery for Refugees interven-
tion and mediated therapeutic effects on PTSD outcomes [20]. A singular qualitative study on
the understanding of self-compassion in Hazara refugees in Australia identified views on self-
compassion that could act as barriers to its application [21]. The existing studies on communi-
ties exposed to trauma does not take into consideration the role of resilience and the complex
interrelationships with mental health.

The link between self-compassion and resilience is complex and only recently being
explored in the literature. Some studies have shown that resilience is promoted by engagement
in self-compassion [17, 22]. In a study of patients with Multiple Sclerosis, the relationship
between self-compassion and health related quality of life is found to be mediated by resilience
[17]. Another study reports that resilience mediates the effect of self-compassion on depression

in a general population sample, but not anxiety [23]. Multiple studies demonstrate that self-compassion is a predictor of resilience [24, 25]. These are promising early indications of the positive impact that the interplay of self-compassion and resilience may have on mental health care and prevention.

Given the lack of resources, high rates of poor mental health outcomes among displaced peoples, and the feasibility of learning self-compassion, this study focuses on the relationship of self-compassion and mental health outcomes amongst Syrian refugees in limited resource settings. Self-compassion is a learned behavior that can help prevent or decrease severity of poor mental health outcomes, especially in lower resource settings with limited access to mental health care. However, there is a complex network between self-compassion, resilience, and mental health. The aims of this study were to investigate the impact of self-compassion on mental health in a population of displaced Syrians living in Jordan, and to explore this effect in relationship to resilience. Specifically, we aimed to assess how levels of self-compassion relate to the prevalence and severity of mental health problems and examine the strength of association between self-compassion and resilience in modulating mental health outcomes. We hypothesized that self-compassion is a strong predictor of poor mental health and positively correlated with resilience. Our study seeks to provide insight into the protective factors that contribute to psychological well-being and provide a path towards novel interventions for mental health care among displaced populations.

## Materials and methods

### Study population

Data was collected through a cross-sectional survey conducted in August 2021 in Amman Jordan. We recruited our target population through local refugee-serving community organizations. Inclusion criteria was being a Syrian refugee adult. A total of 335 participants were recruited through community organizations serving refugees in the Amman area. The research protocol created in partnership with a Jordanian community organization was approved by the University of California, San Diego Institutional Review Board. Community members were invited to the community organization to complete a survey. Community members were informed that completion of the survey was voluntary and that they could withdraw from the study at any time. Written consent was obtained from each participant, and they were subsequently provided with a paper survey in Arabic to complete on their own. Participants received a small gift to thank them for their participation.

Fifteen surveys completed verbally through an interviewer were excluded from the final analysis in order to minimize social desirability bias given the stigma associated with mental health in this community. One survey completed by a participant less than 18 years of age was also excluded from the final analysis. Ten participants did not report their ages on the survey. We did not exclude them and assumed they were above the age of 18.

Surveys received a quality score of 1 = minimal missing data, 2 = missing more than 50% of the data in one section, 3 = missing data in multiple sections. Only surveys with a quality score of 1 were included in the final analysis. In total, 47 surveys with a quality score of 2 or 3 were excluded from the final analysis. A total of 272 surveys were included in the final analysis.

### Inclusivity in global research

Additional information regarding the ethical, cultural, and scientific considerations specific to inclusivity in global research is included in the Supporting Information (S1 Checklist).

## Study measures

Participants were presented the following study measures in the order listed below.

**a. Self-Compassion Scale—Short form (SCS) (Arabic translation).** The SCS is a self-administered 12-item questionnaire measuring the three components of self-compassion including self-kindness, common humanity, and mindfulness [26]. Questions encompass both positive ("When I'm going through a very hard time, I give myself the caring and tenderness I need") and negative ("I'm disapproving and judgmental about my own flaws and inadequacies") aspects of self-compassion. Items are rated on a five-point response scale ranging from 1 (almost never) to 5 (almost always). A total self-compassion score is computed by reversing the negative subscale items and then averaging all subscale scores. The highest score indicates the highest level of self-compassion. Although there are no clinical norms that differentiate between low, moderate, and high self-compassion, a categorization of low SCS = 1.0–2.49, moderate SCS = 2.5–3.5, high SCS = 3.51–5.0 has been suggested in the literature based on population sample means and standard deviations [27]. This categorization is applied here. The translated version has been validated in other Arabic-speaking populations [28]. The scale had a high level of internal consistency, as determined by a Cronbach's alpha of 0.820.

**b. Trauma Experiences Checklist (TEC) (Arabic translation).** The TEC is a self-administered questionnaire, which measures exposure to trauma. It consists of 20 traumatic experiences pertinent to displacement such as "Have you ever had your home forcibly searched by police or armed militia" and "Have you ever seen someone else severely beaten, shot, or killed" to which participants answer "Yes, experienced" or "No, did not experience". The number of items answered "Yes, experienced" is totaled giving a final score. A higher score indicates exposure to more traumatic events. The TEC was created based on the Harvard Trauma Questionnaire and Gaza Traumatic Event Checklist [29]. It has been used in multiple settings involving Arabic-speaking displaced people [30, 31]. The scale had a high level of internal consistency, as determined by a Cronbach's alpha of 0.864.

**c. Hopkins Symptom Checklist (HSCL-25) (Arabic translation).** The HSCL-25 is a symptom inventory that measures adults' symptoms of distress, depression, and anxiety. It consists of 25 items: Part 1 has 10 items for anxiety symptoms and includes statements like "Suddenly scare for no reason". Part II has 15 items for depression symptoms and includes statements like "Feeling low in energy, slowed down". The period of reference is the past month. The scale for each question includes four categories of response ("Not at all", "A little", "Quite a bit", "Extremely" rated 1 to 4, respectively). Three scores are calculated: the emotional distress score is the average of all 25 items, the depression score is the average of the 15 depression items, and the anxiety score is the average of the 10 anxiety items. Total score is highly correlated with severe emotional distress of unspecified diagnosis, and the depression score is correlated with major depression as defined by the DSM-IV based on past studies [32, 33]. A score >1.75 was considered symptomatically positive in our analysis. This cutoff was based on validation in other Arabic-speaking populations and has been used in studies on mental health in Arabic-speaking refugees [34, 35]. We included all three scores in our analysis as suggested by a study validating this scale in Arabic [34]. The scale had a high level of internal consistency for emotional distress, depression, and anxiety subfactors, as determined by Cronbach's alpha's of 0.943, 0.892, and 0.902 respectively.

**d. Connor-Davidson Resilience Scale (CD-RISC) (Arabic translation).** CD-RISC is a self-administered questionnaire of 25 items designed as a Likert type additive scale with five response options (0 = never; 4 = almost always) to statements such as "Sometimes fate or God can help". The final score of the questionnaire is the sum of the responses obtained on each item and the highest scores indicated the highest level of resilience [36]. The translated version

has been validated in other Arabic-speaking populations [37]. The scale had a high level of internal consistency, as determined by a Cronbach's alpha of 0.927.

### Data analysis

Data analysis was conducted using Microsoft Excel and SPSS Statistics Version 29.0.2.0 [38]. Normality of emotional distress, depression, and anxiety across self-compassion groups was assessed by a Shapiro-Wilks test. Mean emotional distress, depression, and anxiety were compared across low, moderate, and high self-compassion groups using Kruskal-Wallis H test. Distributions of distress, depression, and anxiety scores were similar for all groups as determined by visual inspection of boxplots. Post hoc pairwise comparisons were preformed using Dunn's (1964) procedure with a Bonferroni correction. Adjusted p-values are presented. Mean SCS and CD-RISC scores are compared between groups with and without emotional distress, depression, and anxiety using a two-tailed t-test. Homogeneity of variances was confirmed by a Levene's test for equality of variances with $p > 0.05$. We used univariate logistic regression to examine the individual associations of self-compassion, resilience, traumatic experiences, and demographic variables with emotional distress, depression, and anxiety. Multivariate logistic regression was subsequently used to study the combined effects of these variables on emotional distress, depression, and anxiety. A Spearman's rank-order correlation was run to determine the magnitude and direction of the relationship between self-compassion and resilience. Missing values from surveys with a quality score of 1 were conservatively given a value of zero and included in the analysis. All significant values reported have a p-value $< 0.05$.

## Results

### Descriptive analysis

The analysis sample consisted of 272 Syrian participants living in Jordan. Most participants were female (61.8%), married (81.3%), and poorly meeting financial needs (76.8%). The average length of displacement across our sample population was 8 years (SD 2.46). More than half of participants (52.5%) reported having lived in a refugee camp and more than a quarter (27.2%) reported having experienced or witnessed torture (Table 1).

The average total HSCL score, which is highly correlated with emotional distress, was 2.411 (SD 0.625) in our sample. The average HSCL depression score was 2.433 (SD = 0.626) and the average HSCL anxiety score was 2.385 (SD = 0.724). For all HSCL scores, a score >1.75 is considered symptomatic. A majority of those surveyed reported emotional distress (84.6%), depression (85.7%), and/or anxiety (76.5%) (Table1).

The average CD-RISC score was 57.665 (SD = 19.526) which is low relative to the United States (US) general population mean score of 80.7, but comparable to mean scores previously collected among Syrian refugees in Jordan [36, 39]. The SCS average was 3.339 (SD = 0.53) (Table 1). Although there are no clinical norms for the SCS, between 2.5–3.5 can be considered a moderate score [26].

Additionally, a Spearman's rank-order correlation demonstrates a statistically significant moderate positive correlation between self-compassion and resilience in our sample population ($r_s$ = 0.455, p<0.001).

### Comparing emotional distress, depression, and anxiety scores across self-compassion groups

Emotional distress, depression, and anxiety were not normally distributed as determined by a Shapiro-Wilk test with p<0.05. Thus, Kruskal-Wallis H tests were conducted to determine if

**Table 1. Sample population demographic data and mean measure scores stratified by low, moderate, and high SCS score.**

| | N (%)<br>Total N = 272 | Low SCS<br>N (%) | Moderate SCS<br>N (%) | High SCS<br>N (%) |
|---|---|---|---|---|
| **Gender** | | | | |
| Male | 104 (38.2) | 5 (4.8) | 58 (55.7) | 41 (39.4) |
| Female | 168 (61.8) | 11 (6.5) | 102 (60.7) | 55 (32.7) |
| **Age** | | | | |
| 18–30 | 65 (23.9) | 5 (7.7) | 35 (53.8) | 25 (38.4) |
| 31–40 | 86 (31.6) | 7 (8.1) | 53 (61.6) | 26 (30.2) |
| 41–50 | 49 (18.0) | 2 (4.1) | 27 (55.1) | 20 (40.8) |
| 51–60 | 47 (17.3) | 1 (2.1) | 32 (68.1) | 14 (29.8) |
| 61+ | 15 (5.5) | 1 (6.7) | 7 (46.7) | 7 (46.7) |
| Not reported | 10 (3.7) | 0 (0) | 6 (60.0) | 4 (40.0) |
| **Marital Status** | | | | |
| Married | 221 (81.3) | 14 (6.3) | 127 (57.5) | 80 (36.2) |
| Other | 51 (18.8) | 2 (3.9) | 33 (64.7) | 16 (31.4) |
| **Employed** | | | | |
| Yes | 25 (9.2) | 1 (4.0) | 13 (52.0) | 11 (44.0) |
| No | 241 (88.6) | 15 (6.2) | 142 (58.9) | 84 (34.9) |
| Not reported | 6 (2.2) | 0 (0) | 5 (83.3) | 1 (16.7) |
| **Able to meet financial needs** | | | | |
| Poorly | 209 (76.8) | 15 (7.2) | 123 (58.9) | 71 (34.0) |
| Fairly Well | 53 (19.5) | 1 (1.9) | 31 (58.5) | 21 (39.6) |
| Not reported | 10 (3.7) | 0 (0) | 6 (60.0) | 4 (40.0) |
| **Participants with emotional distress** | 230 (84.6) | 16 (7.0) | 142 (61.7) | 73 (31.7) |
| **Participants with depression** | 233 (85.7) | 16 (6.9) | 141 (60.5) | 76 (32.6) |
| **Mean Measure Scores (scale)[a]** | **Mean (SD)** | | | |
| **SCS (1–4)** | 3.339 (0.53) | 2.202 (0.23) | 3.113 (0.25) | 3.906 (0.26) |
| **TEC (0–20)** | 8.699 (4.92) | 9.94 (4.91) | 8.49 (5.08) | 8.84 (4.67) |
| **Emotional Distress (1–4)** | 2.411 (0.625) | 3.10 (0.412) | 2.469 (0.610) | 2.20 (0.577) |
| **Depression (1–4)** | 2.433 (0.626) | 3.195 (0.324) | 2.488 (0.628) | 2.21 (0.538) |
| **Anxiety (1–4)** | 2.385 (0.724) | 2.954 (0.660) | 2.451 (0.678) | 2.181 (0.744) |
| **CD-RISC (0–100)** | 57.665 (19.526) | 41.75 (19.831) | 52.936 (19.057) | 68.198 (15.01) |

Demographic characteristics stratified by low, moderate, and high SCS score reported as N with percentage of sample population in parenthesis unless otherwise indicated. Low SCS = 1.0–2.49, Moderate SCS = 2.5–3.5, High SCS = 3.51–5.0. Abbreviations: SCS, Self-Compassion Scale. TEC, Trauma Exposure Checklist. CD-RISC, Connor-Davidson Resilience Scale.

[a] Scale for measures depicted in parenthesis.

there were differences in emotional distress, depression, and anxiety between those with low, moderate, and high self-compassion. Distributions of emotional distress, depression, and anxiety scores were similar for all groups, as determined by visual inspection of boxplots. The difference in median emotional distress, depression, and anxiety scores were statistically significant between self-compassion groups (emotional distress: $\chi^2(2) = 33.290$, p = <0.001; depression: $\chi^2(2) = 38.785$, p = <0.001; anxiety: $\chi^2(2) = 18.603$, p = <0.001). Pairwise comparisons were preformed using Dunn's (1964) procedure with a Bonferroni correction. Adjusted p-values are presented. This post hoc analysis revealed statistically significant differences in median emotional distress, depression, and anxiety scores across all pairwise self-compassion comparison groups (emotional distress and depression scores: for all pairwise comparisons

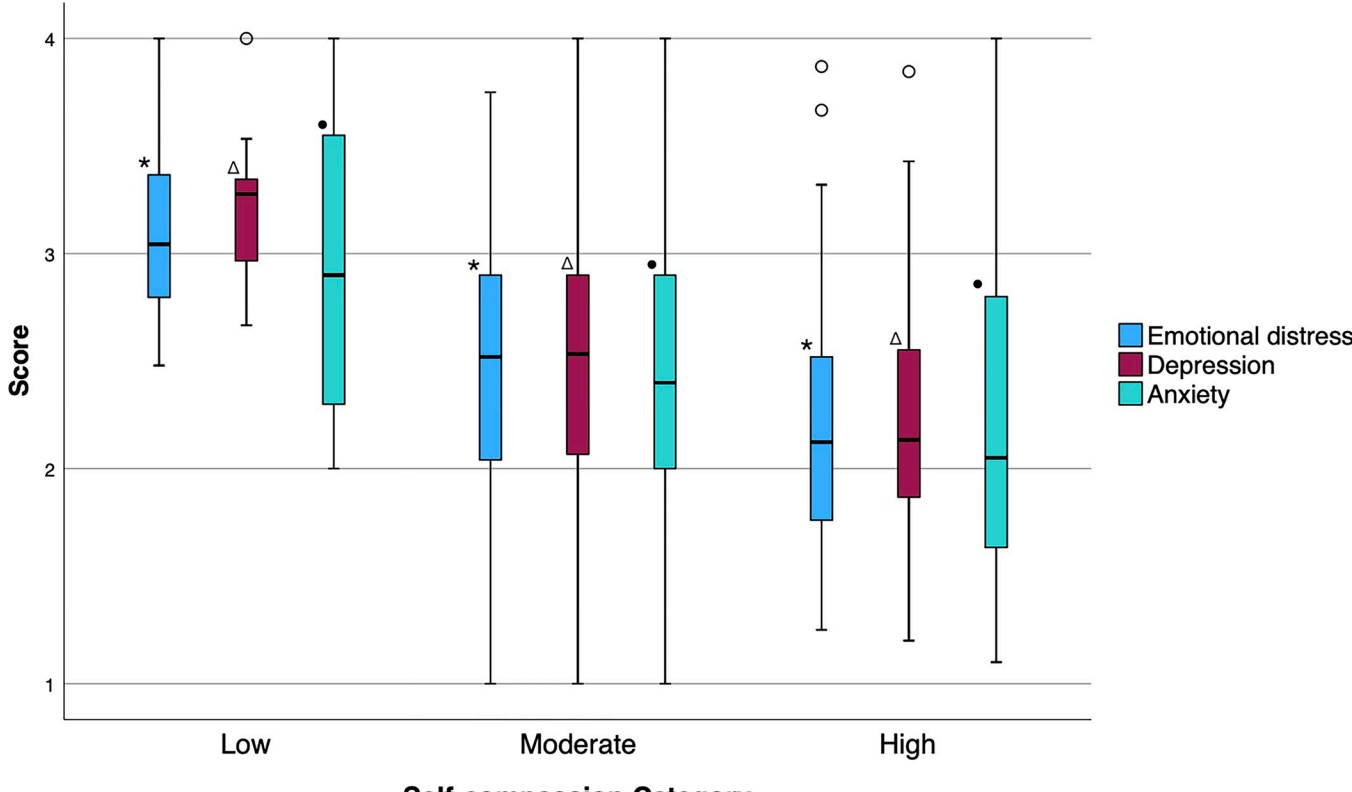

**Fig 1. Emotional distress, depression, and anxiety scores boxplots stratified by low, moderate, and high self-compassion.** Low SCS = 1.0–2.49, Moderate SCS = 2.5–3.5, High SCS = 3.51–5.0. Emotional distress and depression scores greater than 1.75 are considered symptomatic. Within each box the bolded center line denotes the median value; boxes extend from the 25th percentile to the 75th percentile of each group; vertical extending lines denote adjacent values; white circles denote outlier values. *, Δ, and • indicate significant difference between each group as determined by Kruskal-Wallis H test (emotional distress: $\chi^2(2)$ = 33.290, p = <0.001; depression: $\chi^2(2)$ = 38.785, p = <0.001; anxiety: $\chi^2(2)$ = 18.603, p = <0.001) and post hoc pairwise comparison using Dunn's (1964) procedure with a Bonferroni correction (p<0.05). Abbreviation: SCS, Self-Compassion Scale.

p = 0.001, anxiety score: high-moderate p = 0.007, high-low p = 0.001, moderate-low p = 0.043) (Fig 1).

## Comparing self-compassion and resilience scores in presence and absence of emotional distress, depression, and anxiety

Amongst participants with emotional distress, the mean SCS is significantly lower than participants with no emotional distress (t(270) = -4.640, p<0.001) (Table 2). Similarly, participants

**Table 2. Mean SCS and CD-RISC scores stratified by presence or absence of emotional distress, depression, and anxiety.**

| | No Emotional Distress[a] (SE) | Yes Emotional Distress (SE) | p-value[b] | No Depression [a] (SE) | Yes Depression (SE) | p-value [b] | No Anxiety[a] (SE) | Yes Anxiety (SE) | p-value[b] |
|---|---|---|---|---|---|---|---|---|---|
| Mean SCS | 3.693 (0.078) | 3.277 (0.034) | **p<0.001** | 3.643 (0.08) | 3.289 (0.034) | **p<0.001** | 3.666 (0.058) | 3.239 (0.036) | **p<0.001** |
| Mean CD-RISC | 67.488 (3.233) | 55.922 (1.239) | **p<0.001** | 66.333 (3.638) | 56.215 (1.219) | **0.003** | 65.230 (2.436) | 55.340 (1.316) | **p<0.001** |

Abbreviations: SCS, Self-Compassion Scale. CD-RISC, Connor-Davidson Resilience Scale.

[a] Presence of emotional distress, depression, and anxiety determined by HSCL scores >1.75.

[b] P-values from two-tailed t-test displayed.

with depression and participants with anxiety have significantly lower mean self-compassion scores than participants with no depression or anxiety (depression: $t(270) = -3.933$, $p<0.001$; anxiety: $t(270) = -5.933$, $p<0.001$). The mean CD-RISC score is also significantly lower amongst participants with emotional distress ($t(270) = -3.527$, $p<0.001$), depression ($t(270) = -3.040$, $p = 0.003$), and anxiety ($t(270) = -3.625$, $p<0.001$) (Table 2).

## Logistic regression analysis on predictors of emotional distress, anxiety, and depression

In the logistic regression analysis, self-compassion significantly predicted lower emotional distress, depression, and anxiety in both a univariate and multivariate models that included gender, age, marital status, income, occupation, and number of years resettled (Tables 3–5). We also identified gender and ability to meet financial needs as strong predictors of emotional distress, depression, and anxiety. The number of years since displacement was not a predictor of any of these outcomes, however, the number of traumatic experiences was a positive predictor of all three (Tables 3–5). Interestingly, resilience was a predictor of emotional distress, depression, and anxiety in the univariate model. However, this effect became insignificant in the multivariate model for all outcomes (Tables 3–5).

## Discussion

Our study shows for the first time that self-compassion, independent of resilience and other covariates, is predictive of less emotional distress and depression amongst displaced Syrians resettled in Jordan. We demonstrate that self-compassion is, in fact, a more important inverse predictor of poor mental health than resilience in this population. These findings provide insight on how to better support and care for people who are displaced and are at increased risk of poor mental health.

**Table 3. Univariate and multivariate logistic regression analysis on predictors of emotional distress.**

| Variable | Univariate | | | | Multivariate | | | |
|---|---|---|---|---|---|---|---|---|
| | B | OR | 95% CI | P-value | B | OR | 95% CI | p-value |
| **Gender**<br>Male<br>Female (ref)[a] | -1.166 | **0.312** | **(0.158–0.616)** | **<0.001** | -2.943 | **0.053** | **(0.015–0.189)** | **<0.001** |
| **Age** | 0 | 1 | (0.973–1.028) | 0.99 | 0.04 | 1.04 | (0.999–1.083) | 0.055 |
| **Marital Status**<br>Other (single, widowed, divorced)<br>Married (ref) | 0.087 | 1.091 | (0.453–2.629) | 0.846 | 0.312 | 1.366 | (0.371–5.021) | 0.639 |
| **Meeting Financial Needs**<br>Fairly well<br>Poorly (ref) | -1.158 | **0.314** | **(0.153–0.646)** | **0.002** | -1.509 | **0.221** | **(0.080–0.613)** | **0.004** |
| **Currently Employed**<br>Yes<br>No (ref) | -0.587 | 0.556 | (0.208–1.487) | 0.242 | 0.595 | 1.813 | (0.454–7.236) | 0.399 |
| **Number of years resettled** | -0.074 | 0.929 | (0.832–1.038) | 0.192 | -0.061 | 0.941 | (0.829–1.067) | 0.342 |
| **Trauma Exposure Checklist (TEC)** | 0.172 | **1.187** | **(1.095–1.287)** | **<0.001** | 0.313 | **1.367** | **(1.205–1.550)** | **<0.001** |
| **Resilience (CDRISC)** | -0.033 | **0.968** | **(0.950–0.986)** | **<0.001** | -0.014 | 0.986 | (0.962–1.011) | 0.263 |
| **Self-Compassion Scale (SCS)** | -1.589 | **0.204** | **(0.099–0.420)** | **<0.001** | -2.016 | **0.133** | **(0.043–0.415)** | **<0.001** |

[a] (ref) indicates reference category where appropriate.

**Table 4. Univariate and multivariate logistic regression analysis on predictors of depression.**

| Variable | Univariate | | | | Multivariate | | | |
|---|---|---|---|---|---|---|---|---|
| | B | OR | CI | p-value | B | OR | CI | p-value |
| **Gender**<br>Male<br>Female (ref)[a] | -1.005 | **0.366** | **(0.183–0.732)** | **0.004** | -2.679 | **0.067** | **(0.019–0.234)** | **<0.001** |
| **Age** | 0 | 1 | (0.973–1.029) | 0.981 | 0.038 | 1.038 | (0.997–1.082) | 0.071 |
| **Marital Status**<br>Other (single, widowed, divorced)<br>Married (ref) | 0.193 | 1.213 | (0.477–3.084) | 0.684 | 0.61 | 1.84 | (0.478–7.083) | 0.375 |
| **Meeting Financial Needs**<br>Fairly well<br>Poorly (ref) | -1.581 | **0.206** | **(0.099–0.428)** | **<0.001** | -1.978 | **0.138** | **(0.051–0.379)** | **<0.001** |
| **Currently Employed**<br>Yes<br>No (ref) | -0.42 | 0.657 | (0.231–1.868) | 0.431 | 0.924 | 2.521 | (0.598–10.618) | 0.208 |
| **Number of years resettled** | -0.058 | 0.944 | (0.850–1.048) | 0.281 | -0.047 | 0.413 | (0.853–1.068) | 0.413 |
| **Trauma Exposure Checklist (TEC)** | 0.15 | **1.162** | **(1.072–1.259)** | **<0.001** | 0.254 | **1.289** | **(1.148–1.448)** | **<0.001** |
| **Resilience (CDRISC)** | -0.029 | **0.972** | **(0.953–0.990)** | **0.003** | -0.009 | 0.991 | (0.967–1.016) | 0.482 |
| **Self-Compassion Scale (SCS)** | -1.367 | **0.255** | **(0.124–0.522)** | **<0.001** | -1.732 | **0.177** | **(0.059–0.533)** | **0.002** |

[a] (ref) indicates reference category where appropriate.

## Prevalence of emotional distress and depression

Across our study population >75% of participants have symptomatic emotional distress, depression, or anxiety. The rates of mental health morbidity in our study population are higher than those reported in other studies that range between 11–60% [40]. However, most of the existing literature documenting the prevalence of poor mental health in Syrian refugee populations focuses on those resettled in high income countries (HICs), such as Sweden and

**Table 5. Univariate and multivariate logistic regression analysis on predictors of anxiety.**

| Variable | Univariate | | | | Multivariate | | | |
|---|---|---|---|---|---|---|---|---|
| | B | OR | CI | p-value | B | OR | CI | p-value |
| **Gender**<br>Male<br>Female (ref)[a] | -1.412 | **0.244** | **(0.135–0.440)** | **<0.001** | -2.865 | **0.057** | **(0.020–0.161)** | **<0.001** |
| **Age** | -0.024 | 0.977 | (0.954–0.999) | 0.045 | 0.003 | 1.003 | (0.971–1.035) | 0.871 |
| **Marital Status**<br>Other (single, widowed, divorced)<br>Married (ref) | 0.137 | 1.137 | (0.550–2.393) | 0.714 | -0.091 | 0.913 | (0.334–2.494) | 0.859 |
| **Meeting Financial Needs**<br>Fairly well<br>Poorly (ref) | -0.880 | **0.415** | **(0.217–0.792)** | **0.008** | -1.290 | **0.275** | **(0.112–0.677)** | **0.005** |
| **Currently Employed**<br>Yes<br>No (ref) | -0.473 | 0.623 | (0.255–1.520) | 0.298 | 0.534 | 1.706 | (0.496–5.870) | 0.396 |
| **Number of years resettled** | -0.046 | 0.956 | (0.863–1.058) | 0.383 | 0.001 | 1.001 | (0.897–1.117) | 0.990 |
| **Trauma Exposure Checklist (TEC)** | 0.116 | **1.123** | **(1.055–1.195)** | **<0.001** | 0.248 | **1.281** | **(1.164–1.410)** | **<0.001** |
| **Resilience (CDRISC)** | -0.28 | **0.972** | **(0.957–0.988)** | **<0.001** | -0.001 | 0.999 | (0.977–1.020) | 0.908 |
| **Self-Compassion Scale (SCS)** | -1.754 | **0.173** | **(0.090–0.331)** | **<0.001** | -2.187 | **0.112** | **(0.042–0.298)** | **<0.001** |

[a] (ref) indicates reference category where appropriate.

Germany [41–46]. The documented prevalence amongst those resettled in HICs is consistently lower than the prevalence found amongst those living in low- and middle-income countries (LMICs) [35, 47–50]. This might be explained by disparate access to resources in LMICs like Jordan as classified by the World Bank [51]. This is supported by our findings that the ability to meet financial needs is a significant predictor of less emotional distress, depression, and anxiety in both models (Tables 3–5). Other studies conducted in LMICs further support this point. In one study focused on displaced Syrian women attending clinics in Jordan, they report depression affects 62.9% of their sample; anxiety affects 57.5%; and PTSD affects 66.2% [49]. In another study focused on displaced Syrians living in Iraq's Kurdistan Region, utilizing the same measurement tool and positive threshold score applied in our study, the prevalence of depression was 77.2% [35]. Furthermore, our study was conducted in the summer of 2021, 1.5 years into the COVID-19 pandemic. This likely contributed to more prevalent mental health symptoms in our sample population due to the social isolation, fear, and instability from unemployment and pause on resettlements associated with the pandemic.

## Gender, financial status, and traumatic exposure are predictors of mental health

Regression models revealed female gender, poor financial stability, and high traumatic exposure to be persistent predictors of poor mental health. These findings are consistent with the literature on mental health in refugees. Several previous studied have documented that female refugees are more likely to suffer from poor mental health [52, 53]. This is tied to unemployment and weak social networks [53]. Financial instability and history of exposure to trauma are also well-documented predictors of poor mental health outcomes amongst forcibly displaced people [54–56].

## Greater self-compassion is associated with less mental health morbidity

We observed a step-wise decrease in emotional distress, depression, and anxiety across low, moderate, and high self-compassion. Additionally, on average, those with poor mental health had less self-compassion. Furthermore, we found self-compassion to be consistently negatively associated with emotional distress, depression, and anxiety in both univariate and multivariate models. These findings align with existing literature on self-compassion. A meta-analysis of 14 studies from largely HICs measuring the strength of the relationship between self-compassion and psychopathology demonstrates that increased self-compassion is associated with lower levels of mental health symptoms [57]. Some studies have been specifically conducted in populations with traumatic exposure including interpersonal violence and life-threatening illness [14–17]. In one study, self-compassion is identified as a mediating factor between childhood abuse and PTSD [58], and another found that self-compassion modulates the degree of chronicity of PTSD symptoms among veterans [59]. One of the few studies on self-compassion in displaced populations demonstrated that a self-compassion-based intervention improved PTSD outcomes [20]. Another cross-sectional study on the association of self-compassion and depression among Kurdish refugees residing in Norway reports that while self-compassion is associated with depressive symptoms, post-migration work related stressors are a more important predictor of mental health outcomes [19]. Interestingly, in our sample the employment status is not a significant predictor of mental health. However, the employment rate (9.2%) in our sample is much lower than that in Rashidian 2023 (60.4%). Furthermore, both self-compassion and ability to meet financial needs were persistent predictors of mental health in our model. It is possible that self-compassion may moderate the effects of poverty on mental health but not work-related stressors. Further studies are needed to examine the social context in

which self-compassion may be most effective in psychopathology prevention. Our study contributes to the growing body of literature that identifies self-compassion as a predictor of mental health outcomes in the setting of trauma. However, the mechanism by which self-compassion impacts mental health is not completely understood. It is suggested that self-compassion training alters the neural response to evoked pain suggesting that self-compassion has a direct effect on how the brain processes experiences [60]. Importantly, self-compassion has been shown to be modifiable by intervention making its association with improved mental health outcomes even more practically relevant [18, 61, 62]. A randomized control trial evaluating the effectiveness of a mindful self-compassion program found increases in self-compassion being maintained at 6 months and 1 year [18]. The modifiable nature of self-compassion further emphasizes the value of the findings presented in our study to clinicians and systems supporting displaced people.

## Self-compassion is a consistent predictor of mental health independent of resilience

Interestingly, resilience was significantly associated with less emotional distress, depression, and anxiety in our univariate model, but significance was lost in the multivariate model. Resilience is demonstrated in the literature to be protective from negative mental health outcomes [8, 9, 30, 63–68]. However, studies specifically investigating resilience in displaced populations mostly focus on children and adolescents [30, 63–66]. Among the studies that do report on resilience in displaced adults, few utilize direct measures of resilience as we do and rather utilize qualitative approaches [67, 68]. One study reporting on adults displaced from Iraq and living in the US using the CD-RISC measure found that it was associated with less psychological distress but was not a significant predictor of PTSD [8]. Our findings suggest that self-compassion confounded the association of resilience and mental outcomes, and is a more important predictor of mental health outcomes than resilience. The CD-RISC tool measures resilience by evaluating multiple traits that have been described to contribute to resilience including humor, patience, and faith [36]. The complex factors that formulate resilience may introduce variability in predicting mental health outcomes. One longitudinal study examining resilience and mental health after a major life event found that resilience was not predictive of distress over time. In a study of inflammatory bowel disease patients with history of childhood trauma resilience was not a significant moderator between childhood trauma and depression [10]. They suggest that resilience is not innate but influenced by contextual factors such as family relationships.

Our data does demonstrate a moderate positive correlation between self-compassion and resilience that is statistically significant. It is possible that self-compassion alters or interacts with resilience to protect from mental health morbidity, however our findings suggest that self-compassion is the driving factor. Previous studies have linked self-compassion to resilience. In a nonclinical sample of adolescents one study found that self-compassion is positively associated with resilience [69], while another found resilience as a mediating factor between self-compassion and psychological well-being [70]. The relationship between resilience and self-compassion in refugee populations warrants further investigation. Future studies should consider a mediation analysis of these factors in other populations.

## Limitations and future directions

There are a few limitations in our study that should be taken into consideration. We attempted to limit the impacts of social desirability bias by excluding participants who were preliterate from the analysis. However, this may introduce bias from missing information on the role that

self-compassion plays in this subpopulation. Additionally, our study population was recruited through community organizations providing aid and educational opportunities to refugees living in Amman, Jordan, which can exclude less aid dependent Syrians. Conducting surveys at community organizations' offices may have introduced response bias if participants believed they may receive more aid by expressing greater need. However, this should not undermine the internal validity of the results. Additionally, there are potential confounding factors that we did not account for in our analysis. For example, social support or faith practice may be playing a role here. Furthermore, this data was collected in 2021. Although our findings exclude years of resettlement as a predictor of mental health outcomes, it is possible that as we move from a peri-COVID to post-COVID environment the mental health landscape evolves. Therefore, while our study provides valuable insights, ongoing research on mental health is this population is needed. Lastly, the potential for reverse causality from the capability of individuals with less emotional distress, depression, and anxiety to impart self-kindness and understanding cannot be excluded given the cross-sectional design of the study.

Future studies should include randomized clinical trials of self-compassion based interventions in Syrian refugees living in Jordan. Furthermore, qualitative studies should be done to evaluate the receptiveness and cultural barriers to uptake of self-compassion interventions in these communities. A study conducted in Hazara refugees resettled in Australia reported that community views on self-compassion, such as the belief that self-compassion is selfish, may act as barriers to its application [21]. However, this was a small study of only eleven participants. More studies are needed to inform design of self-compassion interventions that are acceptable to displaced peoples.

## Conclusion

Our study is one of the first to demonstrate that self-compassion is inversely predictive of emotional distress, depression, and anxiety independent of resilience in a displaced population. Self-compassion is a modifiable factor that can be utilized by healthcare professionals caring for refugees to promote positive mental health outcomes. Our findings also shed some interesting light on the relationship of resilience, self-compassion, and mental health among displaced vulnerable populations. These areas of positive psychology should be utilized as an addition or alternative, when there are no specialized services, to medical treatment of mild to moderate emotional distress and depression. As the magnitude of displacement increases across the globe due to conflict, climate change, and economic disparities, rates of mental health problems will also increase. Prevention and treatment are major challenges for LMICs, where most displaced populations reside. By better understanding the ways in which communities withstand adversity, we hope to evolve our capabilities to further support vulnerable populations and promote their well-being through efficient and low resource approaches. This is a significant finding that can open the doors for cost-effective interventions without the need for medications or specialized psychiatric care that are lacking in these communities. Randomized controlled trials and longitudinal studies should be implemented to further understanding of the utility of self-compassion for the prevention of poor mental health outcomes in displaced communities.

## Supporting information

**S1 Checklist. Inclusivity in global research checklist.**
(DOCX)

**S1 Data. Data file for open access.**
(XLSX)

# Acknowledgments

The authors would like to thank the participants who generously shared their experiences with us. We are grateful for the efforts of the Jordanian community organizations and their staff that facilitated our engagement with the community.

# Author Contributions

**Conceptualization:** Sarah Alsamman, Rana Dajani, Wael K. Al-Delaimy.

**Data curation:** Sarah Alsamman, Rana Dajani.

**Formal analysis:** Sarah Alsamman, Wael K. Al-Delaimy.

**Methodology:** Wael K. Al-Delaimy.

**Resources:** Wael K. Al-Delaimy.

**Supervision:** Rana Dajani, Wael K. Al-Delaimy.

**Writing – original draft:** Sarah Alsamman.

**Writing – review & editing:** Rana Dajani, Wael K. Al-Delaimy.

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
