## [Decision Letter · Decision Letter 0]

20 Mar 2024

PONE-D-23-39016Self-compassion and association with depression and distress among displaced Syrians: a population-based studyPLOS ONE

Dear Dr. Al-Delaimy,

Thank you for submitting your manuscript to PLOS ONE. After careful consideration, we feel that it has merit but does not fully meet PLOS ONE’s publication criteria as it currently stands. Therefore, we invite you to submit a revised version of the manuscript that addresses the points raised during the review process.

First, I am sorry to take so much time to make a decision and thank you for your patience. Two reviewers withdrew during the editorial process due to their health problems. I have now received one detailed review of your paper, which we have copied below. Of course, I have also thoroughly read the manuscript myself.

As you will see, the reviewer has raised several major issues that you need to address before the paper is considered for publication. Below, I list the major concerns that you will need to address if you decide to revise and resubmit the paper. Additionally, please address all other points raised by the reviewer (find below).

As pointed out by the Reviewer, the introduction can be improved further by going beyond the literature gap and highlighting the novelty and importance of the study.More information is needed in the Materials and Method section. You shared the participants’ information in Table 1, but it would be helpful if you could summarise some general information about the participants in this section. Moreover, it is not clear whether materials were presented in randomised order or the order in the Study Measures section. More information is needed in Study Measures as well. Authors should provide more information on what exactly these materials measure. Providing a couple of sample items can help. Also, the psychometric properties of the scale are missing (see the Reviewer’s comment on this issue).As the reviewer, one of my main concerns is regarding the use of two scores from HSCL-25, namely depression and emotional distress. Why did you choose to use one depression and one emotional distress score from the HSCL-25, instead of one depression and one anxiety or just one emotional distress total? What is the theoretical basis of this decision? What is the statistical basis of this decision? You should clarify this decision both theoretically and statistically (e.g., factor analyses).You should either publish the data publicly available after you anonymised data and omit the sensitive contents or justify your decision to not make data publicly available.You should avoid causal language as this is a cross-sectional survey study. Relatedly, you can tone down your findings to imply that self-compassion is protective against mental health disorders, due to the correlational nature of the study. Self-compassion is definitely associated with less emotional distress; hence, it might be protective against mental health disorders, but it should be further tested in longitudinal or experimental studies.Do you have specific hypotheses you’re testing? Please clearly state that at the end of the introduction.The potential reasons why resilience was significantly associated with less emotional distress and depression in the univariate model, while not in the multivariate model was not discussed thoroughly. I believe authors should further discuss this in the discussion. This finding does not necessarily mean that self-compassion is significantly more predictive of mental health outcomes than resilience. It can be more complex than this, and even if the findings suggest this, the question "why" should be answered. Please see the Reviewer’s points for further expand the discussion.The limitation section is weak. I would like to see improvements concerning the “Limitations and Future Directions”. There is so much to say here about the confounding variables, the social context in which this study was conducted, and the potential of these findings can change from 2011 to 2024, as well as extending future research providing a brief roadmap on this topic.You could directly report the final sample in the abstract to make the abstract shorter and clearer as much as possible.P.3, lines 69-71. Please anonymise your manuscript by omitting information that links the citations to the authors’ details.  In the results section, you should use the name of the concept, instead of the scale. For instance, low to high self-compassion, instead of low to high SCS groups.

We look forward to receiving your revised manuscript.

Kind regards,

Mete Sefa Uysal, Ph.D.

Academic Editor

PLOS ONE

2. Please include a complete copy of PLOS’ questionnaire on inclusivity in global research in your revised manuscript. Our policy for research in this area aims to improve transparency in the reporting of research performed outside of researchers’ own country or community. The policy applies to researchers who have travelled to a different country to conduct research, research with Indigenous populations or their lands, and research on cultural artefacts. The questionnaire can also be requested at the journal’s discretion for any other submissions, even if these conditions are not met.  Please find more information on the policy and a link to download a blank copy of the questionnaire here: https://journals.plos.org/plosone/s/best-practices-in-research-reporting. Please upload a completed version of your questionnaire as Supporting Information when you resubmit your manuscript

4. In this instance it seems there may be acceptable restrictions in place that prevent the public sharing of your minimal data. However, in line with our goal of ensuring long-term data availability to all interested researchers, PLOS’ Data Policy states that authors cannot be the sole named individuals responsible for ensuring data access (http://journals.plos.org/plosone/s/data-availability#loc-acceptable-data-sharing-methods).

Reviewers' comments: 

Regarding the article, I have found significant strengths, especially for the findings of the study. In addition, the study may address the predictive roles of resilience and self-compassion for mental health outcomes in a disadvantaged population. However, there are some issues that may limit the potential contributions of this study. I hope that the authors will find helpful the following issues that I pointed. 

**Introduction**

The introduction of the manuscript requires improvement. One of my main concern is the lack of clarity regarding the novelty of the study, what this study adds novelty in relation to the existing literature. It is recommened to emphasize the relationship between resilience, self-compassion, and mental health outcomes, rather than simply saying that no study has examined these variables among refugees-this is not sufficient to establish novelty-. Another concern is the limited knowledge about the related variables and functions for traumatized individuals. Please, include the existing literature to understand the association of these variables in the scope of mental health. Lastly, please clarify the aim of the study.

Line.73 Please explain what you mean “work upstream”

Line.80-82. Self-compassion has been worked with refugees in contemporary research. Please check the relevant sentence and the related articles Rashidian et al. (2023), Ghasemy (2020) etc

Line.82-83. *“Furthermore, the link between self-compassion and resilience is explored in the literature, but not defined in the context of displaced populations**”*

How these two factors, self-compassion and resilience, are associated in other sample as you stated in this sentence? Please, add more information.

**Materials and Methods**

Consider reorganizing this section by adding subtitles under methods and materials, such as participants, procedure, statistical analysis etc. Please refer to the journal archieves for guidance on how to organize this part.

Did you have any inclusion or exclusion criteria for sample selection in your study? If you had, please indicate.

Line.102-105. You excluded one participants as being younger than 18 years old. However, when I examined the Table 1, I noticed that the age section had a  “not reported” category. How did you determine that these 10 participants were older than 18 or younger than 80?

Line. 102-105. How did you detect social desirability in your sample? Did you use any validated measurement tool for identifying it?

Line. 106-109. I assume what you mean by “the quality scores” is missing values in dataset. If not, could you explain how you determined the quality of the data. I suggest that you provide statistical evidence why you excluded these data.

Measurement tools:

You employed Arabic version of all the scales. I wonder about the psychometric properties of these scales. Do the scales have adequate validity and realibility scores for Arabic individuals? In addition, how about the internal consistency scores of your sample for all measurement tools?

Line.118-128. Indeed, the HSCL-25 has two sub-dimensions: anxiety and depression. According to the references you cite (18 and 19) researchers have also calculated anxiety and depression seperately. The Arabic version (Fares et al., 2021) recommended bifactor model: two factors (anxiety/depression) and a thir factor (underlying 25-items). However, you calculated a total score for emotional distress and 15 items for depression separately- it was not calculated any score for anxiety-.The introduction part mentioned that anxiety is also common adverse mental health outcomes among refugees. I wonder why you did not measure anxiety for your sample?

I assume you have used the information for HSCL-25 and calculation of its from this website https://hprt-cambridge.org/screening/hopkins-symptom-checklist. I recommend that you cite this website in your manuscript or other articles which point the same total score and depression sub-dimension. Additionally, please paraphrase the sentences, as there is a high degree of similarity between your sentences and the text on this website.

Line.133 “This will be a secondary exposure to self-compassion measures” I did not understand what you meant with this sentence. Please explain.

Line.144-151. To make this section easier to read and avoid confusion, I suggest titling it as “statistical analysis”

This section requires additional information regarding the statistics that the authors used. Were there any missing values? As I read previous paragraph that you wrote (for participation selection), some individuals did not complete all items in the data set. Thus, please include information about missing value. If any missing values, how were missing values handled. There is some assumptions for conducting ANOVA. I wonder whether these assumptions were checked or not. Specifically, was the normality of the data checked? Did data distribute normally? Please check the assumptions of ANOVA and explain it in a clear way in the text.

Line.144.Which version of SPSS was used? All the statistics that you mentioned could be carried out in SPSS? Why did you use excel?

Line.144-146. How were the low, moderate, and high self-compassion scores of the participants determined? I knew that this scale does not have any cut-off scores. (If wrong, please inform me and add the text)

Line.146 “…. ANOVA followed by post hoc Tukey’s Honest Significance Test with Bonferroni correction”. Did you use Tukey or Bonferroni? I am confused.

Line.148. Please rewrite the sentence in a clearer way. For instance, we used univariate logistic regression for ….. and multivariate for ….. etc.

Results

In general, results section is a kind of complicated. It is hard to understand and follow the findings. I advise the authors adding subtitle under results section to make easier the interpretation for instance descriptive analysis, prevalence, logistic regression analysis -for emotional distress, for depression etc. This will be helping to increase understandability.

Line.174-176. Please, explain why did you conduct correlation? If you have a hypothesis, please indicate in the text or you may think to exclude it.

Line.203-211. As I suggested above, maybe you can add distinct subheadings for emotional distress and depression.

Discussion

In general, it is recommended that the authors may present their finding at first, and then discuss them based on the existing literature, providing possible explanations of their findings. Secondly, the authors discussed only findings of resilience and self-compassion. However, gender, meeting financial needs, and TEC were also found to be significant predictors of emotional distress and depression in logistic regression. Please, include a discussion about these findings in this part. Lastly, what does this study suggest for future studies or clinical implications? Please, provide your suggestions in a separate paragraph.

Please indicate that Jordan is one of the LMICs according to (World Bank?) …. Since you interpreted migration to LMICs could be an explanation.

Line.241 The authors specifically emphasized the term “severely traumatized…” in some sentences. When you said that, I expected an objective indicator that provides the trauma level of your sample or your sample just consisted of individuals with high levels of traumatic stress. I believe that simply stating this term will not enough to prove all of the sample experienced severe trauma. Maybe instead of emphasizing the term so much, it may be sufficient to briefly introduce about the trauma that experienced by Syrian refugees in the introduction part.

Line.249-250. As I stated previously, there have been some studies conducted on self-compassion among refugees. Please check it and add.

“***Greater self-compassion is associated with less mental health morbidity***”

First, you can present your findings of this study. Then, you can discuss how your finding placed in the existing literature. The psychosocial intervention findings that you cited may be incorporated into clinical or future implications by combining them with your findings.

**“Self-compassion is a consistent predictor of lower emotional distress and depression independent of resilience**
**”**

Please explain the finding of why resilience did not predict emotional distress and depression based on existing literature. You cited an article numbered 6, how they explain this finding? Moreover, some researchers argued that being resilient may not be associated with a reduction in distress (e.g., Blanke et al., 2023).

Line.274-283. The authors discussed this correlation finding. However, as previously stated, you should determine the aim of the manuscript and if you wish to include this finding, please clarify the aim of the study in introduction part.

Reviewer's Responses to Questions

1. Is the manuscript technically sound, and do the data support the conclusions?

Reviewer #1: Yes

2. Has the statistical analysis been performed appropriately and rigorously? 

Reviewer #1: Yes

3. Have the authors made all data underlying the findings in their manuscript fully available?

Reviewer #1: No

4. Is the manuscript presented in an intelligible fashion and written in standard English?

Reviewer #1: Yes

5. Review Comments to the Author

Reviewer #1: The article highlights an intriguing topic and presents some notable findings for future clinical implications. However, there are some issues that should be improved. Introduction needs to be revised to include some additional knowledge based on the existing literature. The methods should be clarified, especially in terms of the statistical analysis and measurement tools that were used. The Results section should be reorganized. Lastly, the discussion part should be improved to interpret the findings of the current study and some other variables that were found significant predictors should be interpreted.

6. PLOS authors have the option to publish the peer review history of their article (what does this mean?). If published, this will include your full peer review and any attached files.

Reviewer #1: No

---

## [Author Response · Author response to Decision Letter 0]

8 Jun 2024

Response to Reviewer comments 

Regarding the article, I have found significant strengths, especially for the findings of the study. In addition, the study may address the predictive roles of resilience and self-compassion for mental health outcomes in a disadvantaged population. However, there are some issues that may limit the potential contributions of this study. I hope that the authors will find helpful the following issues that I pointed. 

Response: 

Thank you for your comments. We have incorporated your feedback into the manuscript and believe it has stregthened the presentation of our findings. 

Introduction 

1) The introduction of the manuscript requires improvement. One of my main concern is the lack of clarity regarding the novelty of the study, what this study adds novelty in relation to the existing literature. It is recommened to emphasize the relationship between resilience, self-compassion, and mental health outcomes, rather than simply saying that no study has examined these variables among refugees-this is not sufficient to establish novelty-. Another concern is the limited knowledge about the related variables and functions for traumatized individuals. Please, include the existing literature to understand the association of these variables in the scope of mental health. Lastly, please clarify the aim of the study. 

Response: 

The introduction was edited to calirfy the novelty of the study. Given the increase in displaced populations globally and the limited resources for mental health care, interventions of self-compassion can address this gap among the most vulnerable populaitons. We belive that this study presents valuable data regarding the relevance of self compassion as a protective factor from mental health morbidity in refugees independent of resilience. This is important given the high rates of mental health disorders in communities that have experienced forced displacement and the need for tailored interventions. 

As suggested by the reviewer, we have added literature and text to the introduction to highlight the association between resilience and mental health and trauma, as well as the association between self-compassion and trauma. 

We have also clarified the aim of the study in the last part of the introduction. To summarize the aims of this study were to investigate the impact of self-compassion on mental health disorders in a population of displaced Syrians living in Jordan, and to explore this effect in relationship to resilience.

2) Line.73 Please explain what you mean “work upstream”.

Response: 

This phrasing was edited to clarify the meaning. Resilience is a complex multidimensional process that encompasses multiple subfactors. Interventions that improve resilience may act on specific subfactors like self-comapssion rather than attempt to modulate resilience directly.

3) Line.80-82. Self-compassion has been worked with refugees in contemporary research. Please check the relevant sentence and the related articles Rashidian et al. (2023), Ghasemy (2020) etc 

Response:

Thank you for sharing these articles. We edited the introduction to include these and other studies on self-compassion in refugees. 

4) Line.82-83. “Furthermore, the link between self-compassion and resilience is explored in the literature, but not defined in the context of displaced populations” 

How these two factors, self-compassion and resilience, are associated in other sample as you stated in this sentence? Please, add more information. 

Response: 

We removed this sentence and provided more literature on the association between self-compassion and resilience. 

Materials and Methods 

5) Consider reorganizing this section by adding subtitles under methods and materials, such as participants, procedure, statistical analysis etc. Please refer to the journal archieves for guidance on how to organize this part. 

Response:

Thank you for this suggestions. We added subtitles for clarity. 

6) Did you have any inclusion or exclusion criteria for sample selection in your study? If you had, please indicate. 

Response:

We did not have any exclusion criteria. Any Syrian refugee adult was included. We have added this to the Methods. However, during analysis we excluded those who participated and did not have completed surveys. 

7) Line.102-105. You excluded one participants as being younger than 18 years old. However, when I examined the Table 1, I noticed that the age section had a “not reported” category. How did you determine that these 10 participants were older than 18 or younger than 80? 

Response:

The reviewer raises a good point. We removed the age criteria because we included any adult 18 years or older. We do not know if the 10 participants who did not report their age included someone below the age of 18. We have added a sentence to the methods to clarify this. 

8) Line. 102-105. How did you detect social desirability in your sample? Did you use any validated measurement tool for identifying it? 

Response:

We did not use a social desirability measurement tool, however, after the interviews were completed the authors concluded that respondents were avoiding questions about mental health. This is likely because the setting was not conducive of discussing these stigmatized topics through interview and that is why we had self-report surveys for collection of data given the associated stigma and the inavailabilty of private office space during the data collection. We thus preferred to exclude those who were preliterate and had to be interviewed verbally in order to avoid potential bias in the results. 

9) Line. 106-109. I assume what you mean by “the quality scores” is missing values in dataset. If not, could you explain how you determined the quality of the data. I suggest that you provide statistical evidence why you excluded these data. 

Response:

We clarified in this sentence that these scores were related to data completion. 

Measurement tools

10) You employed Arabic version of all the scales. I wonder about the psychometric properties of these scales. Do the scales have adequate validity and realibility scores for Arabic individuals? In addition, how about the internal consistency scores of your sample for all measurement tools? 

Response:

We did not embark on psychometric analysis of these well validated scales because we made the assumption that the reliability scores would not change based on the language. However, the reviewer raises a valid point that can be addressed in future psychometric focused analyses and papers but that is beyond the scope of this paper. We did report on the internal consistency of each measure by including a Cronbach’s alpha all of which were greater than 0.8. Thank you for this suggestion. 

11) Line.118-128. Indeed, the HSCL-25 has two sub-dimensions: anxiety and depression. According to the references you cite (18 and 19) researchers have also calculated anxiety and depression seperately. The Arabic version (Fares et al., 2021) recommended bifactor model: two factors (anxiety/depression) and a thir factor (underlying 25-items). However, you calculated a total score for emotional distress and 15 items for depression separately- it was not calculated any score for anxiety-.The introduction part mentioned that anxiety is also common adverse mental health outcomes among refugees. I wonder why you did not measure anxiety for your sample? 

I assume you have used the information for HSCL-25 and calculation of its from this website https://hprt-cambridge.org/screening/hopkins-symptom-checklist. I recommend that you cite this website in your manuscript or other articles which point the same total score and depression sub-dimension. Additionally, please paraphrase the sentences, as there is a high degree of similarity between your sentences and the text on this website.

Response:

The reviewer brings up an important point here and we agree that anxiety should be included in the anlaysis. The updated manuscript now reports anxiety as an outcome in all analyses. 

Additionaly, we cited the HPRT website and edited the text describing the HSCL-25 to reduce similarity with source text. 

12) Line.133 “This will be a secondary exposure to self-compassion measures” I did not understand what you meant with this sentence. Please explain.

Response:

Thank you for pointing this out to us, we have removed this sentence. 

13) Line.144-151. To make this section easier to read and avoid confusion, I suggest titling it as “statistical analysis”

Response:

Thank you for this suggestion. We included this subtitle. 

14) This section requires additional information regarding the statistics that the authors used. Were there any missing values? As I read previous paragraph that you wrote (for participation selection), some individuals did not complete all items in the data set. Thus, please include information about missing value. If any missing values, how were missing values handled. There is some assumptions for conducting ANOVA. I wonder whether these assumptions were checked or not. Specifically, was the normality of the data checked? Did data distribute normally? Please check the assumptions of ANOVA and explain it in a clear way in the text. 

Response:

The reviwer raises multiple excellent points here. Missing values were conservatively assigned a value of zero and were included in the analysis. We added this information to the Statistical Analysis section. 

With the inclusion of anxiety in our analysis ANOVA assmuptions were not met, we therefore decided to conduct the nonparametric Kruskal-Wallis H test. The distribution of distress, depression, and anxiety scores were determined similar by visual inspection of boxplots. Post hoc pairwise comparisons were preformed using Dunn’s (1964) procedure with a Bonferroni correction. This is now clearly described in the text. 

15) Line.144.Which version of SPSS was used? All the statistics that you mentioned could be carried out in SPSS? Why did you use excel? 

Response:

SPSS Version 29.0.2.0 was used. We added this to the text and included a citation for SPSS. Because data was originally stored in Excel it was utilized for early calculations o the descriptive statistics. All other analysis was conducted in SPSS. 

16) Line.144-146. How were the low, moderate, and high self-compassion scores of the participants determined? I knew that this scale does not have any cut-off scores. (If wrong, please inform me and add the text)

Response:

The reviewer is correct in that there are no clinical norms which differentiate between low, moderate, and high self compassion. However, we used an ad hoc rubric that has been suggested in the literature based on score means and standard deviations in population samples. We added information on this to the SCS scale description under Study Measures. More information on this can be found here: 

https://self-compassion.org/wp-content/uploads/2022/01/Self-CompassionScaleChapter.pdf

17) Line.146 “…. ANOVA followed by post hoc Tukey’s Honest Significance Test with Bonferroni correction”. Did you use Tukey or Bonferroni? I am confused. 

Response:

We apologize for the confusion this sentence has been removed from the manuscript. 

18) Line.148. Please rewrite the sentence in a clearer way. For instance, we used univariate logistic regression for ….. and multivariate for ….. etc. 

Response:

Thank you for this suggestion, we agree the original text was unclear. We edited the text to clarify. 

Results 

19) In general, results section is a kind of complicated. It is hard to understand and follow the findings. I advise the authors adding subtitle under results section to make easier the interpretation for instance descriptive analysis, prevalence, logistic regression analysis -for emotional distress, for depression etc. This will be helping to increase understandability. 

Response:

Thank you for this suggestion. We added subtitles to clarify the results section. 

20) Line.174-176. Please, explain why did you conduct correlation? If you have a hypothesis, please indicate in the text or you may think to exclude it. 

Response:

We were interested in the magnitude of correlation between resilience and self compassion given the theoretical framework suggesting a relationship between these constructs in this population. We added a review of this literature to the introduction. Understanding the correlation between self-compassion and resilience also informed our statistical modeling. For example, if resilience and self compassion were highly related they may have led to multicollinearity issues and made the regression model unstable if both were included. The hypothesis is that resilience and self compassion are positively correlated. We edited our aims and hypothesis in the Introduction section to clarify these points. 

21) Line.203-211. As I suggested above, maybe you can add distinct subheadings for emotional distress and depression. 

Response:

We added subtitles to the results section to make it easier to read. We decided not to split up the summary of logistic regression results by emotional distress, depression, and anxiety in order to avoid being repetitive given similarity of results across the three outcomes. 

Discussion 

22) In general, it is recommended that the authors may present their finding at first, and then discuss them based on the existing literature, providing possible explanations of their findings. Secondly, the authors discussed only findings of resilience and self-compassion. However, gender, meeting financial needs, and TEC were also found to be significant predictors of emotional distress and depression in logistic regression. Please, include a discussion about these findings in this part. Lastly, what does this study suggest for future studies or clinical implications? Please, provide your suggestions in a separate paragraph. 

Response:

Thank you for your suggestions. We incorporated them into our disucssion and overall we believe it has strengthened it. Specifcally, we added a discssuion of implications and suggestions for future studies. We also address gender, financial needs and TEC as predictors of emotional distress, depression, and anxiety. 

 23) Please indicate that Jordan is one of the LMICs according to (World Bank?) …. Since you interpreted migration to LMICs could be an explanation. 

Response:

Thank you for pointing this out. We specified in the text that Jordan is classified as a LMIC by the World Bank and included a citation. 

24) Line.241 The authors specifically emphasized the term “severely traumatized…” in some sentences. When you said that, I expected an objective indicator that provides the trauma level of your sample or your sample just consisted of individuals with high levels of traumatic stress. I believe that simply stating this term will not enough to prove all of the sample experienced severe trauma. Maybe instead of emphasizing the term so much, it may be sufficient to briefly introduce about the trauma that experienced by Syrian refugees in the introduction part. 

Response:

The reviewer brings up a good point. We removed the term “severely” from the manuscript. 

25) Line.249-250. As I stated previously, there have been some studies conducted on self-compassion among refugees. Please check it and add. 

Response:

Thank you again for brining these papers to our attention. We added a discussion of these studies to the manuscript. 

26) “Greater self-compassion is associated with less mental health morbidity”

First, you can present your findings of this study. Then, you can discuss how your finding placed in the existing literature. The psychosocial intervention findings that you cited may be incorporated into clinical or future implications by combining them with your findings.

Response:

We edited this section to present our findings first and then discuss in the context of existing literature. Thank you for this suggestion. 

27) “Self-compassion is a consistent predictor of lower emotional distress and depression independent of resilience”

Please explain the findin

---

## [Decision Letter · Decision Letter 1]

19 Jul 2024

PONE-D-23-39016R1Self-compassion and association with distress, depression, and anxiety among displaced Syrians: a population-based studyPLOS ONE

Dear Dr. Al-Delaimy,

Thank you for submitting your manuscript to PLOS ONE. After careful consideration, we feel that it has merit but does not fully meet PLOS ONE’s publication criteria as it currently stands. Therefore, we invite you to submit a revised version of the manuscript that addresses the points raised during the review process.

In addition to a reviewer who reviewed the initial submission, I read the manuscript thoroughly. Please carefully address the reviewer's point (see below), as they are crucial, particularly regarding the statistical considerations, although they are very straightforward and could be easily addressed. In addition to the reviewer's point, I have two minor, but important, points: 

- Abstract: Please simplify the abstract removing the statistical results such as p values and OR, and focusing on the summary of your research findings and contributions (please also see the reviewer's comment on the abstract)

- I suggest avoiding terms like “mental illnesses” or “mental health disorders” (for instance see the first paragraph in the introduction, lines 59-60, or page 5., lines 104 and 111; please check it throughout the manuscript), as they might refer clinical settings and not appropriate for non-hospitalised vulnerable groups like refugees. You could adopt the terms such as mental health problems instead.

We look forward to receiving your revised manuscript.

Kind regards,

Mete Sefa Uysal, Ph.D.

Academic Editor

PLOS ONE

Journal Requirements:

Reviewers' comments: 

Regarding the revision of the manuscript, I believe that this version exhibits greater strength than the previous version. I would like to appreciate the authors for their efforts.

**In the abstract**

In the abstract, depending on your findings, you might consider adding the predictive role of gender, financial status and traumatic exposure The incorporation of these variables, you might think about making abstract more attractive and interesting.

**In Methods and Results sections**

Lines 198-199. In the study population section, you wrote:  “In total, 47 surveys with a quality score of 2 or 3 were excluded from the final analysis.”. However, in this sentence you wrote “Missing values were conservatively given a value of zero and included in the analysis”. As before, you stated you exclude the data with missing values. Please, check it.

You applied non parametric test with the inclusion of anxiety. Regarding this information, I understand that the anxiety measurement was not normally distributed but what about other measurement such as depression, emotional distress? If only anxiety was non-normally distributed, I recommend using Kruskal-Wallis H test for anxiety, one-way ANOVA for other measurements if normal distrubiton was met.

Reviewer's Responses to Questions

**Comments to the Author**

1. If the authors have adequately addressed your comments raised in a previous round of review and you feel that this manuscript is now acceptable for publication, you may indicate that here to bypass the “Comments to the Author” section, enter your conflict of interest statement in the “Confidential to Editor” section, and submit your "Accept" recommendation.

Reviewer #1: All comments have been addressed

2. Is the manuscript technically sound, and do the data support the conclusions?

Reviewer #1: Yes

3. Has the statistical analysis been performed appropriately and rigorously? 

Reviewer #1: Yes

4. Have the authors made all data underlying the findings in their manuscript fully available?

Reviewer #1: Yes

5. Is the manuscript presented in an intelligible fashion and written in standard English?

Reviewer #1: Yes

6. Review Comments to the Author

Reviewer #1: (No Response)

7. PLOS authors have the option to publish the peer review history of their article (what does this mean?). If published, this will include your full peer review and any attached files.

Reviewer #1: No

---

## [Author Response · Author response to Decision Letter 1]

29 Jul 2024

Self-compassion and association with distress, depression, and anxiety among displaced Syrians: a population-based study

-Regarding the revision of the manuscript, I believe that this version exhibits greater strength than the previous version. I would like to appreciate the authors for their efforts. 

Response: 

Thank you for the all thoughtful feedback you have provided which has ultimately strengthened this paper. 

In the abstract 

-In the abstract, depending on your findings, you might consider adding the predictive role of gender, financial status and traumatic exposure The incorporation of these variables, you might think about making abstract more attractive and interesting.

Response: 

We included this finding in the abstract. Thank you for this suggestion. 

In Methods and Results sections

-Lines 198-199. In the study population section, you wrote: “In total, 47 surveys with a quality score of 2 or 3 were excluded from the final analysis.”. However, in this sentence you wrote “Missing values were conservatively given a value of zero and included in the analysis”. As before, you stated you exclude the data with missing values. Please, check it. 

Response: 

Thank you for bringing this to our attention. We edited the text to further clarify the quality score assigned to each survery. To summarize, surveys received a quality score of if they had minimal missing data, 2 if they were missing more than 50% of the data in one section, and 3 if they were missing data in multiple sections. Missing values from surveys with a quality score of 1 were then conservatively given a value of zero. 

-You applied non parametric test with the inclusion of anxiety. Regarding this information, I understand that the anxiety measurement was not normally distributed but what about other measurement such as depression, emotional distress? If only anxiety was non-normally distributed, I recommend using Kruskal-Wallis H test for anxiety, one-way ANOVA for other measurements if normal distrubiton was met. 

Response: 

Thank you for this comment. Depression, emotional distress, and anxiety were not normally distrubted across self compassion groups which is why Kruskal-Wallis H test was ultimatley applied. Absence of normality was determined by a Shapiro-Wilk test with p < 0.05. 

Editor Comments:

-Abstract: Please simplify the abstract removing the statistical results such as p values and OR, and focusing on the summary of your research findings and contributions (please also see the reviewer's comment on the abstract)

Response: 

We removed the statistical results from the abstract and included more of our findings per the reviewers suggestion (ie gender, financial status, and trauma exposure). 

-I suggest avoiding terms like “mental illnesses” or “mental health disorders” (for instance see the first paragraph in the introduction, lines 59-60, or page 5., lines 104 and 111; please check it throughout the manuscript), as they might refer clinical settings and not appropriate for non-hospitalised vulnerable groups like refugees. You could adopt the terms such as mental health problems instead.

Response: 

Thank you for this suggestion. The terms “mental illnessses” and “mental health disorders” have been removed from the manuscript and replaced with “poor mental health” or “mental health problems”.

---

## [Editor Report · Decision Letter 2]

6 Aug 2024

Self-compassion and association with distress, depression, and anxiety among displaced Syrians: a population-based study

PONE-D-23-39016R2

Dear Dr. Al-Delaimy,

We’re pleased to inform you that your manuscript has been judged scientifically suitable for publication and will be formally accepted for publication once it meets all outstanding technical requirements.

I have a minor suggestion that does not require another round of revision, but you can update it in the amendment and proofreading period. I recommend updating the following sentence in the abstract for clarity: “Female gender, poor financial stability, and high levels of traumatic exposure were also identified as persistent predictors of mental health morbidity.” You may prefer something like “Gender (i.e, females have poorer mental health compared to males), poor financial stability, and high levels of traumatic exposure were also identified as predictors of mental health problems.” 

Kind regards,

Mete Sefa Uysal, Ph.D.

Academic Editor

PLOS ONE

---

## [Editor Report · Acceptance letter]

11 Sep 2024

PONE-D-23-39016R2 

PLOS ONE

Dear Dr. Al-Delaimy, 

I'm pleased to inform you that your manuscript has been deemed suitable for publication in PLOS ONE. Congratulations! Your manuscript is now being handed over to our production team.

Kind regards, 

on behalf of

Dr. Mete Sefa Uysal 

Academic Editor

PLOS ONE